# Fluctuations and Selection Bias in 5 and 13 TeV
# *p-p* Collisions: Where are the jets?

**Thomas A. Trainor⋆**

University of Washington, Seattle, USA

⋆ [ttrainor99@gmail.com](mailto:ttrainor99@gmail.com)

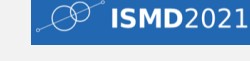

*50th International Symposium on Multiparticle Dynamics*
*(ISMD2021)*
*12-16 July 2021*
doi:[10.21468/SciPostPhysProc.10](https://doi.org/10.21468/SciPostPhysProc.10)

## Abstract

The ALICE collaboration recently reported high-statistics $p_t$ spectra from 5 TeV and 13 TeV *p-p* collisions with intent to determine the role of jets in high-multiplicity collisions. In the present study a two-component (soft + hard) model (TCM) of hadron production in *p-p* collisions is applied to ALICE $p_t$ spectra. As in previous TCM studies of A-B collision systems jet and nonjet contributions to $p_t$ spectra are accurately separated over the entire $p_t$ acceptance. The statistical significance of data-model differences is established leading to insights concerning selection bias and spectrum model validity.

doi:[10.21468/SciPostPhysProc.10.003](https://doi.org/10.21468/SciPostPhysProc.10.003)

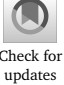 Check for updates

## 1 Introduction

A trend has emerged to interpret certain phenomena commonly associated with "collectivity" in A-A collisions as evidence for QGP formation in smaller collision systems if the same phenomena appear there. To address claims of collectivity in *p-p* collisions the present study applies a two-component (soft + hard) model (TCM) of hadron production near midrapidity [1, 2] to evaluate jet production in 13 TeV *p-p* $p_t$ spectra [3]. The study reveals how jets contribute to spectra, especially at lower $p_t$, and what selection biases result from event sorting according to two different $\eta$ acceptances. Z-scores are employed to determine the statistical significance of data-model differences. Based on Z-scores four models are evaluated: fixed TCM, variable TCM, Tsallis model and blast-wave model. Spectrum data appear to be inconsistent with collectivity (flows) or jet modification.

## 2 Published 13 TeV *p-p* $p_t$ spectra vs TCM reference

The ALICE collaboration recently published high-statistics $p_t$ spectra for 5 and 13 TeV *p-p* collisions [3]. The stated purpose was "...to investigate the importance of jets in high-multiplicity pp collisions and their contribution to charged-particle production at low $p_T$."

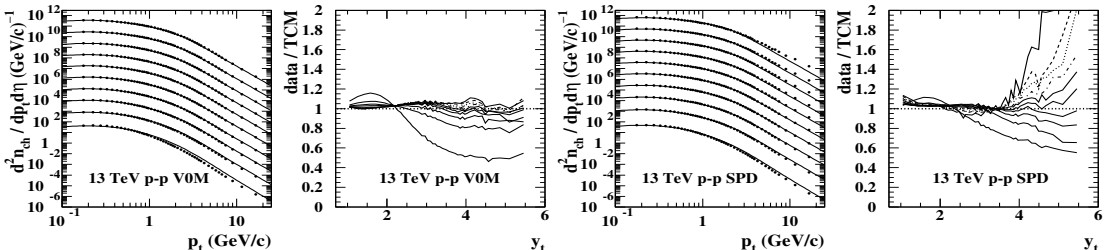

Figure 1: Published $p_t$ spectra (points) and TCM (solid) for 13 TeV *p-p* collisions and for V0M (left) and SPD (right) event selection together with data/TCM data-model ratios.

Figure 1 shows published 13 TeV spectra (points) and fixed two-component model (TCM) for two methods (V0M and SPD $\eta$ acceptances) of sorting the same event (and jet) ensemble. The TCM provides excellent descriptions of data at lower $p_t$, but systematic deviations depending on the selection method (V0M or SPD $\eta$ acceptance) arise at higher $p_t$ and for *lower* $n_{ch}$. The TCM is then used as a fixed reference to study selection bias.

## 3 Two-component Model – Jets vs Nonjet Production

The two-component (soft + hard) model (TCM) of hadron production near midrapidity is a predictive model that can be applied to any A-B collision system [4–7]. Its simple algebraic structure is described by (with $\bar{\rho}_x \equiv n_x/\Delta\eta$)

$$\bar{\rho}_0(y_t; n_{ch}) \approx \frac{d^2 n_{ch}}{y_t d y_t d\eta} \approx \bar{\rho}_s \hat{S}_0(y_t) + \bar{\rho}_h \hat{H}_0(y_t), \tag{1}$$

where transverse rapidity $y_{ti} \equiv \ln[(m_{ti} + p_t)/m_i]$ is defined for hadron species $i$ and $\hat{S}_0(y_t)$ and $\hat{H}_0(y_t)$ are unit-normal fixed model functions. The model is based on (a) factorization of $n_{ch}$ and $y_t$ dependence for each component and (b) the critical relation $\bar{\rho}_h \approx \alpha(\sqrt{s})\bar{\rho}_s^2$ with $\alpha(\sqrt{s}) \approx O(0.01)$, both features originally inferred from spectrum data [4]. The soft component is interpreted to represent projectile-nucleon dissociation along the *p-p* collision axis. The hard component is interpreted to represent large-angle scattered low-$x$ gluons fragmenting to dijets. The direct relation between spectrum hard components and measured jet properties has previously been demonstrated quantitatively [8,9].

Figure 2 illustrates decomposition of measured $p_t$ spectra into soft and hard components. Normalized spectra $X(y_t)$ as defined by the first relation in Eqs. (2) are shown in first and third panels. The normalized spectra all coincide with TCM model $\hat{S}_0(y_t)$ (bold dashed) within data uncertainties below 0.5 GeV/c ($y_t \approx 2$). The hard/soft density ratio is defined by $x(n_s) \equiv \bar{\rho}_h/\bar{\rho}_s \approx \alpha\bar{\rho}_s$.

$$X(y_t) \equiv \frac{\bar{\rho}_0(y_t)}{\bar{\rho}_s} \approx \hat{S}_0(y_t) + x(n_s)\hat{H}_0(y_t); \quad Y(y_t) \equiv \frac{1}{x(n_s)}\left[X(y_t) - \hat{S}_0(y_t)\right] \approx \hat{H}_0(y_t). \tag{2}$$

Spectrum hard components $Y(y_t)$ as defined by the second relation in Eqs. (2) are shown in second and fourth panels compared to model function $\hat{H}_0(y_t)$ (bold dashed). The inferred

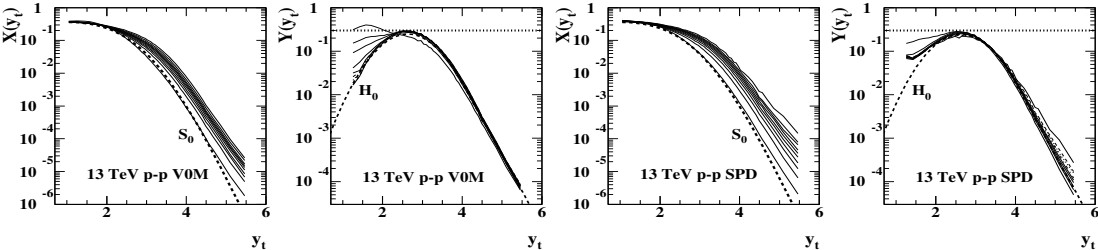

Figure 2: Normalized $p_t$ spectra $X(y_t)$ and inferred jet-related hard components $Y(y_t)$ for V0M (left) and SPD (right) event selection. $\hat{S}_0(y_t)$ and $\hat{H}_0(y_t)$ are fixed model functions.

spectrum hard components accurately represent the full jet contribution to spectra over the *entire $p_t$ acceptance* subject to certain biases depending on event selection criteria [1,2].

## 4 Measures of Spectrum Structure and Bias Trends

Reference [3] introduces several methods to assess jet contributions to spectra. One is plots of data spectra in ratio to a common reference data spectrum. Another is fitting a power-law model to data spectra to infer exponent $n$ possibly related to a jet contribution.

Figure 3 (left) shows spectrum ratios $X_i(y_t)/X_{ref}(y_t)$ [see Eq. (2)] where $X_{ref}(y_t)$ corresponds in this example to event class 5. Even with the normalized form $X(y_t)$ such spectrum ratios confuse two remaining issues: jet production and selection bias. Contrast with Fig. 1 (second and fourth) where selection bias can be studied in isolation.

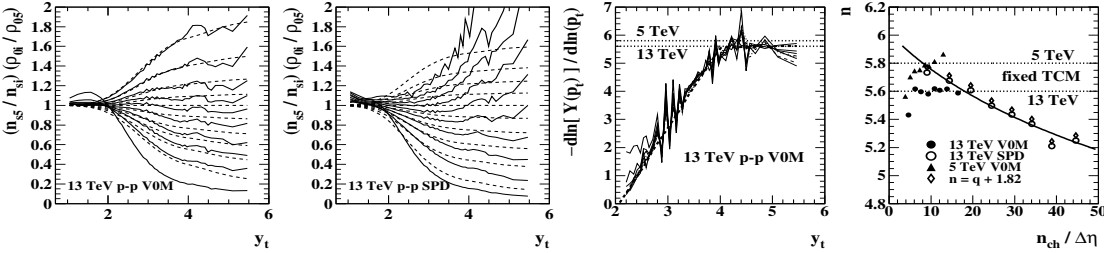

Figure 3: Left: Normalized $p_t$ spectra $X_i(y_t)$ in ratio to reference spectrum $X_5(y_t)$. Right: Hard-component logarithmic derivatives and power-law exponents $n$ for V0M and SPD.

A power-law exponent $n$ can be inferred directly from spectrum data in a model-independent way (no power-law model fits required) via the logarithmic derivative

$$-\frac{d\ln[Y(p_t)]}{d\ln(p_t)} \to n \ \text{ for large } p_t, \tag{3}$$

where $Y(p_t) = y_t Y(y_t)/m_t p_t$. The result for data hard component $Y(p_t)$ must be the same as for intact data spectra because soft component $\hat{S}_0(p_t)$ is negligible at relevant $p_t$ values.

Figure 3 (third) shows log derivatives for V0M spectrum hard components. Above $y_t = 4$ ($p_t \approx 4$ GeV/c) inferred $n$ values stabilize near $n = 5.6$ (for 13 TeV). Figure 3 (fourth) shows the major difference between V0M (fixed $n$) and SPD (strongly varying $n$) trends: different responses to fluctuating parton fragmentation from different $\eta$ intervals.

# 5  Statistical Significance and Z-scores

In comparisons of data with models it is essential to determine the statistical significance of data-model deviations. The appropriate measure is the Z-score [10]. The data/model *ratio* is a common method of data-model comparison which however typically conveys a misleading representation of model quality. The essential relations are conveyed by

$$\frac{\text{data}}{\text{model}} - 1 \approx \frac{\text{data} - \text{model}}{\text{error}} \times \frac{\text{error}}{\text{data}}, \tag{4}$$

where the approximation arises from replacing error/model by error/data. The first factor at right defines Z-scores as data-model *differences* in ratio to statistical uncertainties.

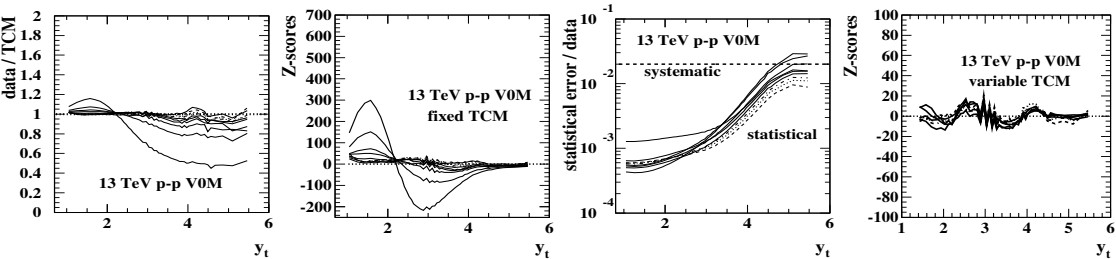

Figure 4: First: Data/model ratios for V0M selection. Second: Corresponding Z-scores. Third: Error/data ratios. Fourth: Z-scores for *variable* TCM in relation to V0M data.

Figure 4 (first) shows V0M data/model ratios as in Fig. 1 (second). The second panel shows the corresponding Z-scores while the third panel shows the error/data ratios for V0M spectra corresponding to 60 million *p-p* events. The fourth panel shows Z-scores for a *variable* TCM in which the hard-component model is adjusted to accommodate data (solely by varying the model width below its mode). Surviving residuals are common to all event classes and both event selection methods suggesting the presence of an artifact introduced during data processing. Note the matching noise structure in Fig. 3 (third).

# 6  Alternative Spectrum Models

Certain models have been applied to $p_t$ spectra from small collision systems in connection with claims of collectivity in those systems, for instance Tsallis and blast-wave (BW) models. Those models were applied recently to 13 TeV *p-p* spectra as reported in Ref. [11].

## 6.1  Tsallis model – extent of equilibration

Figure 5 (first) shows Tsallis model fits (solid) compared with V0M spectra. The second and third panels show corresponding data/model ratios and Z-scores. The fourth panel shows Z-scores for the variable TCM on the same vertical scale. The relation between the $\chi^2$ statistic and Z-scores is $\chi^2 = \sum_i Z_i^2$. Thus, $\chi^2$ values for the Tsallis model are in this case $O(10,000)$ [2]. The Tsallis model is dramatically rejected by spectrum data.

## 6.2  Blast-wave model – hydrodynamic flows

Figure 6 (first) shows BW model fits (solid) to the same V0M spectra. The second and third panels show corresponding data/model ratios and Z-scores. Again one encounters $\chi^2$ values

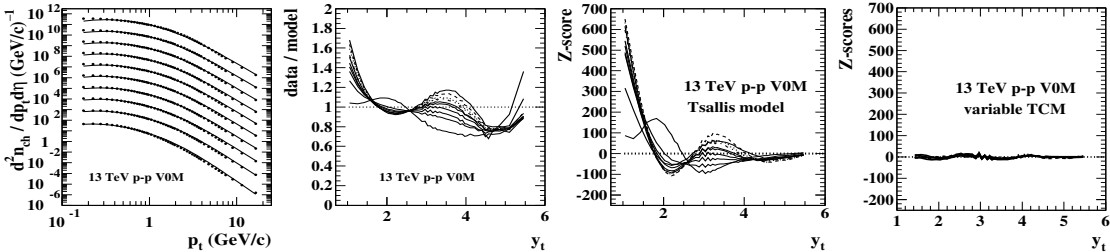

Figure 5: First: Tsallis model (solid) vs V0M spectra (points). Second: Tsallis data/model ratios. Third: Corresponding Z-scores. Fourth: Z-scores for variable TCM.

$O(10,000)$ for the BW model. The fourth panel shows the same BW model applied to 5 TeV $p$-Pb data compared to a TCM for identified hadrons reported in Ref. [6]. It is notable that whereas the BW model (dashed) misses the $p$-Pb data to the same degree as in the first panel the TCM for $K_S^0$ (solid) describes the data within their uncertainties from 7 GeV/c *down to zero*, precluding any possibility of a radial-flow contribution [2].

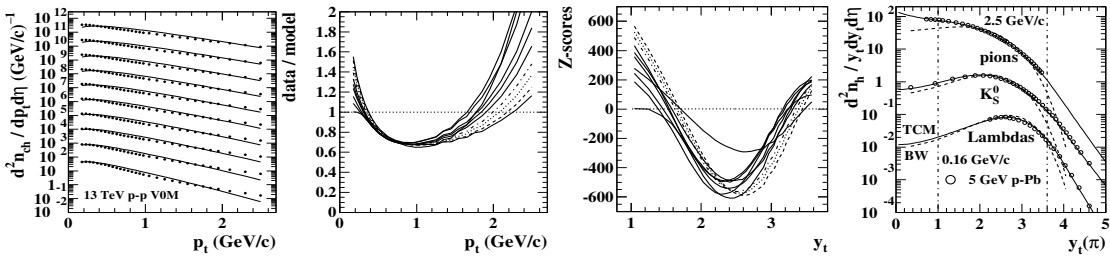

Figure 6: First: Blast-wave model (solid) vs V0M spectra (points). Second: Data/model ratios. Third: Corresponding Z-scores. Fourth: BW model applied to 5 TeV $p$-Pb spectra.

## 7 Conclusion

The same ensemble of 60 million 13 TeV $p$-$p$ collisions is partitioned in two ways (V0M vs SPD). A two-component model (TCM) facilitates precise separation of $p_t$ spectra into nonjet (soft) and jet-related (hard) components over the entire $p_t$ acceptance. Complementary biases of the jet-related hard component relative to the *fixed* TCM as reference are observed reflecting the relation of different $\eta$ acceptances to jet production mechanisms. A *variable* TCM (variation of hard-component model widths below *or* above the mode) is adjusted to accommodate data. Z-scores (data-model deviations in ratio to statistical uncertainties) are introduced to evaluate deviation *significance*. Z-scores are then used to test model validity. The variable TCM is observed to describe spectra within their point-to-point uncertainties whereas Tsallis and blast-wave models, often invoked in the context of "collectivity" and incomplete thermal equilibration, are dramatically falsified with Z-scores equivalent to $\chi^2$ values $O(10,000)$. Careful examination of spectrum evolution with charge-multiplicity and pseudorapidity conditions reveals no significant evidence for radial flow or for jet modification that might buttress claims of $p$-$p$ collectivity.

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
