# Peer review of "Fluctuations and Selection Bias in 5 and 13 TeV p-p Collisions: Where are the jets?"

_SciPost Physics Proceedings, doi:SciPost Phys. Proc. 10, 003 (2022)_

## Round 1 · Referee Report · Anonymous (Referee 1) · 2022-1-28

Strengths

A coherent and mostly clear write-up of this long-running interest of the author, and I feel I understand better as a result.

Weaknesses

The actual connection between the H component of the yt spectrum and other exclusive event quantities isn't made clear. Jets are an empirically existent local phenomena in events, so presumably the statistical scaling can be expected to break down at some point. Do the events compared to have selections requiring a certain number of jets in a particular algorithm above a certain pT, and how does that event-selection affect the S and H functions?

Report

This proceedings contribution describes the state of investigations in the inclusive TCM very well and is suitable for publication as-is.

In future studies and reports I would be very interested to see how this "inclusive scaling" approach to highly active collider events interplays with more local requirements on jets in event selection. Hope we can do that in this year's ISMD!

---

## Editorial Decision

published